# Recovering Complete Actions for Cross-dataset Skeleton Action Recognition

**Hanchao Liu**[1]    **Yujiang Li**[1]    **Tai-Jiang Mu**[1*]    **Shi-Min Hu**[1]

[1]BNRist, Department of Computer Science and Technology, Tsinghua University

## Abstract

Despite huge progress in skeleton-based action recognition, its generalizability to different domains remains a challenging issue. In this paper, to solve the skeleton action generalization problem, we present a recover-and-resample augmentation framework based on a novel complete action prior. We observe that human daily actions are confronted with temporal mismatch across different datasets, as they are usually partial observations of their complete action sequences. By recovering complete actions and resampling from these full sequences, we can generate strong augmentations for unseen domains. At the same time, we discover the nature of general action completeness within large datasets, indicated by the per-frame diversity over time. This allows us to exploit two assets of transferable knowledge that can be shared across action samples and be helpful for action completion: boundary poses for determining the action start, and linear temporal transforms for capturing global action patterns. Therefore, we formulate the recovering stage as a two-step stochastic action completion with boundary pose-conditioned extrapolation followed by smooth linear transforms. Both the boundary poses and linear transforms can be efficiently learned from the whole dataset via clustering. We validate our approach on a cross-dataset setting with three skeleton action datasets, outperforming other domain generalization approaches by a considerable margin.

## 1 Introduction

Skeleton-based action recognition has recently achieved great success [57, 42, 6, 29]. The skeleton-based action representation has the advantage of removed background changes and camera positions, making it more robust compared to RGB representation. However, the generalizability to unseen domains under such a representation is still affected by the inherent spatiotemporal difference of 3D coordinates of a same action across domains, which yet largely remains an under-explored issue. In this paper, we study the single domain generalization problem [55] for skeleton-based action recognition, in which we do not have knowledge about the target domain.

Essentially, both cross-subject and cross-view settings [41] fall under the category of cross-domain settings, and they can be well addressed by designing more powerful backbones and applying geometric transformations, achieving high accuracy in the test set [6, 29]. However, we find that in the *cross-dataset* setting where source and target data come from different datasets, the performance on accuracy degrades a lot (around or more than 20% in some cases [46]) and cannot be well remedied by the above approaches. This indicates drastic domain gap in the inherent feature of human actions across datasets, posing great challenges for real-life applications [40, 18] and calling for research on domain generalization techniques for skeleton-based representation, which is the focus of this paper.

---

*Tai-Jiang Mu is the corresponding author. Email: taijiang@tsinghua.edu.cn. Code is available at `https://github.com/HanchaoLiu/Recover-and-Resample`

38th Conference on Neural Information Processing Systems (NeurIPS 2024).

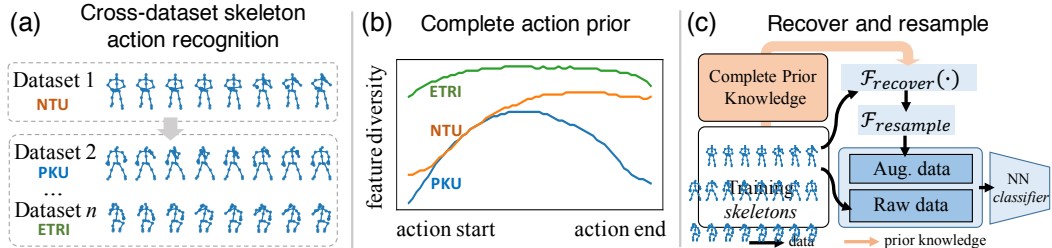

Figure 1: (a) **Cross-dataset skeleton action recognition.** Taking action *phone calling* as an example, temporal mismatch across datasets poses a challenging issue. (b) **Complete action prior.** Human actions within large datasets exhibit statistical patterns from less feature diversity to more diversity, implying the nature of action completeness (shown for NTU, PKU and ETRI dataset). (c) **Recover and Resample.** After learning a stochastic action completion function from the training data, we recover complete actions and resample from them to further augment the training set.

Investigating action samples across multiple datasets, our observation is that a notable source of domain gap comes from the temporal mismatch of an action across different datasets (Fig. 1(a)), which is usually caused by different definition or cropping criterion of human motions. Existing methods cannot perfectly handle this problem. Warping-based temporal alignment approaches, while being popular for few-shot recognition tasks [3, 53, 33, 45], are inefficient and not optimal when given sufficient and diverse training data. Besides general domain generalization approaches [35, 51], self-supervised skeleton representation learning methods [23, 47, 46, 59], while showing good transferability of the learned feature, are still difficult to handle data with large gaps as the generalizability largely comes from seeing handcrafted augmentations in either contrastive learning [62] or auxiliary tasks [46]. Different from the above approaches, we aim to directly hallucinate strong augmentations for unseen domains by exploring some sort of human action priors.

As in Fig. 1 (b), we train a skeleton auto-encoder and examine the per-frame latent feature diversity of the whole dataset (measured by standard deviation) for several skeleton action datasets, i.e., NTU [41], PKU [25], ETRI [15] (Details in Appendix A5). We observe that human action sequences start with relatively low feature diversity, which is actually a form that humans perform generally *complete* actions within large datasets, from rest poses that are less diverse (e.g. stand, sit) to rich-semantic poses that are more diverse. We summarize this pattern as a novel temporal prior named **complete action prior**. Although this prior can be detected by statistics in a general sense, in terms of individual samples, some exhibit strong action completeness while some are segments of their complete actions. This motivates us to learn an action completion function from the whole training data which can transform incomplete actions into complete ones. In this way, we can recover the complete sequence of an action from its partial observation, and resample from it to hallucinate data in unseen domains.

Going deeper into such a statistical pattern, we can further mine some class-agnostic knowledge from the whole dataset that can be used for our action completion. First, the low diversity at beginning frames implies the existence of a set of representative boundary poses, which can be used for determining the start of an action. Second, the general completeness implies the existence of long and nearly complete actions within a large dataset. By studying the relationships between their raw and trimmed pairs, we can learn temporal patterns inherited in human actions.

Based on the above observations, we propose a novel recover-and-resample augmentation framework for single domain generalization on skeleton action recognition. As in Fig. 1 (c), we recover complete actions from training samples and resample from them to further augment the training set. Especially, in the recovering stage, we adopt a two-step stochastic action completion, which first extrapolates the raw sequence conditioning on the boundary pose and then applies temporal transforms. A set of boundary poses are learned from the first frames of the training data. Altogether, a set of smooth linear transforms (linearity means re-organizing existing frames) are learned from reconstructing full sequence from its trimmed segment via a closed-form solution of context-aware frame similarity aggregation. We find such a simplified form of transform suitable and expressive enough for modeling common structural temporal patterns, e.g., shifting, scaling, symmetry, etc. The learning for both boundary poses and linear transforms can be achieved via clustering, for example, $k$-means, which makes our whole framework light and efficient. Also, we find a random cropping suffices for the resampling stage.

We simplify the problem in cross-domain settings by studying well-defined daily actions in indoor settings. Specifically, we construct a new evaluation setting jointly with three large-scale datasets including PKU-MMD [25], NTU-RGBD [41] and ETRI-Activity3D [15]. Our proposed augmentation method well solves the temporal misalignment issue. We improve the average accuracy on unseen datasets by 5%, outperforming other baseline methods by a large margin.

In summary, the contributions of this paper are as follows:

- We discover the complete action prior within large datasets by its statistical pattern, whose effect has not been well studied before. Building on such a prior, we present a novel recover-and-resample augmentation framework for domain generalization on skeleton-based action recognition.
- We propose an effective clustering-based approach to recover complete actions, which is achieved by boundary pose conditioned extrapolation and context-aware smooth linear transforms.
- We demonstrate the superiority of our method over a wide range of methods by a large margin and we conduct extensive experiments to show the effectiveness of each proposed module.

## 2 Related Work

### 2.1 Skeleton-based Action Recognition

As for skeleton action recognition, the graph neural network [57, 42, 6, 29] has become a prevailing model due to its effectiveness to model spatiotemporal relations between joints. We use AGCN (Adaptive GCN) [42] as our backbone as it can model flexible graph layouts. Recently, there is a focus on studying skeleton action recognition with less labeled data. For self-supervised skeleton representation learning, a number of pretext tasks are proposed based on invariant augmentations [23, 47, 11, 59, 56]. These approaches improve the generalizability across datasets but are still bounded by the specific augmentation designs. For few-shot skeleton action recognition, metric learning approaches with different types of dynamic time warping [3, 53, 33, 45] are widely adopted. However, they are not optimal for domain generalization given large amount of training data. Our work especially explores the generalization problem for this field when data is not available in the target domain.

### 2.2 Domain Generalization and Data Augmentation

General domain generalization approaches mainly rely on learning adversarial augmentation [51, 22, 55] and self-supervised auxiliary tasks [4]. They are not optimal for skeleton-based action recognition tasks since they do not make full use of the skeleton representation. The recent ST-Cubism [46] is most related to ours, which adapts learning jigsaw puzzles [46] to skeleton sequences to achieve domain generalization. While there are many augmentation methods intended for non-semantic related tasks like pose estimation [16, 2, 10, 9, 14] and human motion prediction [31], skeleton augmentation for recognition tasks is essentially challenging as it is fragile to spoil the semantics of skeletons formed by low-dimensional joints. SFN [34] combines Mixup [61] with VAE for augmentation, but it still cannot deal with cross-domain settings. ModSelect [32] proposes a selection mechanism for multi-modal input for cross-domain action generalization. In this work we focus on the learnable temporal action augmentation for cross-domain settings.

### 2.3 Human Motion Priors

As human motions are constrained in the spatiotemporal space, many human motion priors are proposed, such as manifolds of valid human poses [19, 7, 48] and motion [63, 27, 17], and other properties such as periodical human motion pattern [43, 12, 44], body part interchangeability across samples [26, 24] and alignment with other modalities [65, 1]. ACTOR [38] learns a CVAE and samples data points conditioning on labels as an augmentation method for recognition tasks. However, it is only useful when data is scarce. Rate-invariant prior [50] is related to ours, but changing rate itself is not enough for cross-domain settings. In this paper, we explore the action completeness prior for better performance in cross-domain settings. Although the notion of action completeness [30, 64] appears in the temporal action localization (TAL) task [60], we are the first to incorporate it for the skeleton-based domain generalization task.

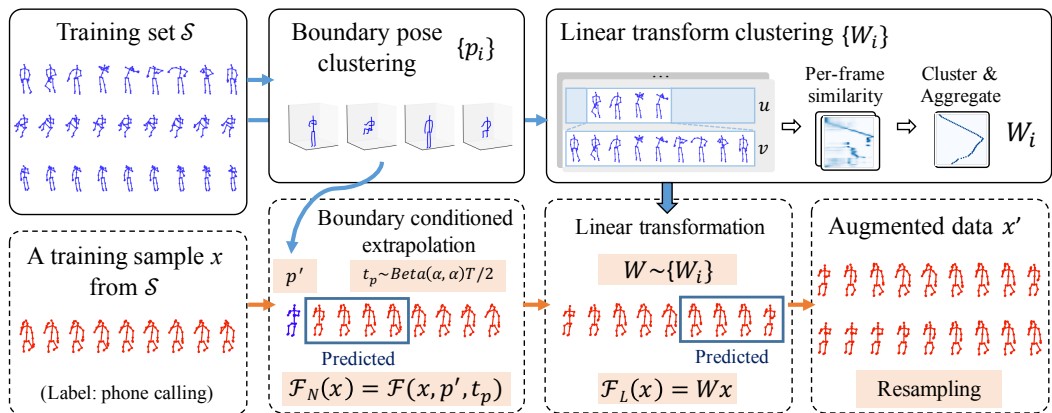

Figure 2: **Overview of Recovering and Resampling.** Given training set $\mathcal{S}$, we learn boundary poses $\{p_i\}$ and context-aware linear transforms $\{W_i\}$ via clustering. For a sample $x$ from $\mathcal{S}$, we first do **extrapolation** ($\mathcal{F}_\mathcal{N}$) conditioning on the boundary pose $p'$ with infilling length $t_p$, and then perform **linear transform** ($\mathcal{F}_\mathcal{L}$) by sampling from $\{W_i\}$. The new data points $x'$ are **resampled** from recovered complete actions as strong augmentations for unseen datasets. Skeletons in dark blue rectangles are new frames generated by $\mathcal{F}_\mathcal{N}$ and $\mathcal{F}_\mathcal{L}$. Both $x$ and $x'$ are used for training the classifier.

## 3  Method

### 3.1  Problem Setting

In the skeleton-based action recognition, suppose we have source data from one single domain $\mathcal{S} = \{(x_\mathcal{S}, y_\mathcal{S})\}$. Due to the unfixed length of input action sequences, the skeleton sequences $x_\mathcal{S}$ are all uniformly resized to $x_\mathcal{S} \in \mathbb{R}^{T \times J \times 3}$ with sequence length $T$ and joint number $J$ [6, 15]. Our goal is to train a model $\mathcal{G}$ using $\mathcal{S}$ and test on other unseen target domains $\{(x_{\mathcal{T}1}, y_{\mathcal{T}1}), (x_{\mathcal{T}2}, y_{\mathcal{T}2}), \cdots \}$. In particular $y_\mathcal{S}$ and $y_\mathcal{T}$ share the same action categories.

### 3.2  A Recover-and-Resample Framework

Motivated by the observations of temporal mismatch (Fig. 1(a)) and action completeness within a dataset (Fig. 1(b)), we first introduce a generalized form of temporal augmentation, which we name recover-and-resample augmentation. Given a training sample $x$, we first recover its complete action with a transform $W_\text{recover}$ and then sample a segment of it using $W_\text{resample}$ as an augmentation $x'$:

$$x' = W_\text{full}(x) = (W_\text{resample} \circ W_\text{recover})(x). \tag{1}$$

Ideally $W_\text{full}$ can generate all kinds of temporal segments that may appear in the target domains. In contrastive self-supervised learning, [23] uses temporal shift as a recovering process, while [47] only uses resampling for constructing positive samples. We set the resample transform $W_\text{resample}$ as a common random sampling [47], and focus more on the recovering stage $W_\text{recover}$ that can construct a complete action for the input motion $x$. Our contribution is that by exploiting the knowledge from complete action prior, we further decompose $W_\text{recover}$ into a linear transform and a nonlinear transform, i.e. $W_\text{recover}(x) = \mathcal{F}_L(\mathcal{F}_N(x))$. The nonlinear transform $\mathcal{F}_N$ extrapolates the motion to generate new poses which do not exist in the original motion. The linear transform $\mathcal{F}_L$ reorganizes the motion sequence using existing frames. In this way we boost the expressive power of $W_\text{recover}$ for generating complete actions. Fig. 2 gives an illustration on how we generate augmentations for a partially observed action *phone calling*. Details for $\mathcal{F}_L$ and $\mathcal{F}_N$ are introduced in Sec. 3.3 and 3.4.

### 3.3  Boundary-conditioned Extrapolation

**Boundary pose clustering.** When recovering a complete action, we first determine its boundary pose. Intuitively, boundary poses for common daily activities are more likely to be *background* poses (or rest poses). They do not convey strong action semantics and can be distinguished from meaningful foreground poses. On the other hand, the statistical finding that the initial poses of skeletal

action sequences have low feature diversity (shown in Fig. 1 (b)) also validates our assumption: the boundary poses are to some extent constrained. Therefore, we propose to cluster the first frames $x_0$ of all the training samples in $\mathcal{S}$ to get a set of representative background (boundary) poses, which we denote as $\{p_i\}$. During clustering, all the first frames $\{x_0 | x_0 \in \mathbb{R}^{J \times 3}\}$ are flattened to 1d vectors and $L_2$ norm is used as the distance measure.

**Conditional generation.** We use nearest neighbour search to assign the boundary pose $p'$ for $x$ with first frame $x_0$:

$$p' = \arg\min_{\{p_i\}} |x_0 - p_i|. \tag{2}$$

Now we can extrapolate the original first frame $x_0$ to the new boundary pose $p'$. This is basically an infilling process conditioned on $p'$, and we control the length of extrapolation with parameter $t_p$. Supposing the sequence length is $T$, we set the first frame as $p'$, and squeeze the original motion $x$ to the segment from frame $t_p$ to the end frame $T$. We then infill the timestamps from the first frame to frame $t_p$ with new motion. Considering a stochastic process, we sample $t_p$ from a Beta distribution $\beta(\alpha, \alpha)T/2$ with high probability on the first frame and frame $T/2$. This means the conditional generation process is more likely to retain the original motion or infill a segment with length $T/2$.

**Motion infiller.** The simplest form of the motion infiller $\mathcal{F}$ is linear interpolation since the skeleton already has good joint correspondence. We also propose an alternative to train an infiller with a neural network. We mask out some consecutive frames to get masked and full sequence pairs and train an action completion network. We then use this motion infiller to perform extrapolation. The comparison will be shown in the later experiments. Actually we find the parameter-free linear interpolation works well enough for generating reasonable motions for the recognition task.

So formally our nonlinear transform can be represented as $\mathcal{F}_N(x) = \mathcal{F}(x, p', t_p)$. We first find the boundary pose $p'$, sample $t_p$ to determine the sequence length to infill, and finally apply the motion infiller $\mathcal{F}$.

### 3.4 Learning Smooth Linear Transforms

Note that the above nonlinear transform is still unable to capture global and structural patterns inherited in the human actions. Consequently, we further propose linear transforms $\mathcal{F}_L$ to reorganize frames for more powerful transformations. We have seen great progress in self-supervised learning with very simple temporal linear transforms like *crop & pad* [23]. However they are manually designed and are not learned from data so they are not flexible. Here we propose to construct partial and full skeleton sequence pairs to learn a set of linear transforms in the form of $\mathcal{F}_L(x) = Wx, W \in \mathbb{R}^{T \times T}$. Here $x \in \mathbb{R}^{T \times J \times 3}$ is a skeleton sequence with length $T$.

**Context-aware smooth linear transforms.** Given partial sequence $u$ and full sequence $v$ (see Fig. 2) which are then both resized to length $T$, we hope to find a linear transform that minimizes the $L_2$ norm $|Wu - v|$, where $W$ has only one nonzero element for each row. The straightforward solution is to do it frame-wise, i.e. for each frame $v_i$ in $v$ we find the index of its closest frame $u_j$ in $u$. However, such an operation cannot ensure that $W$ is smooth and consistent with good transferability. Inspired by the context-aware alignment [20], we propose *context-aware smooth linear transform* that aggregates information from semantically adjacent frames.

Specifically, we first find the context-aware frame-wise similarity matrix of $u$ and $v$. For example, the similarity score $s_{ij}$ between *i*-th frame of $v$ and *j*-th frame of $u$ can be obtained as following:

$$s_{ij} = \frac{\exp(-|v_i - u_j|/\lambda_T)}{\sum_{m=1}^{T} \exp(-|v_i - u_m|/\lambda_T)}, \tag{3}$$

where $|\cdot|$ denotes $L_2$ distance between two flattened skeletons (root translation already removed). Larger $\lambda_T$ means stronger context-aware smoothing. We then calculate the index by:

$$k_i = \sum_{j=1}^{T} j \cdot s_{ij}. \tag{4}$$

The transform matrix $W$ is then obtained by setting $W[i, round(k_i)] = 1$. This is to say, when deciding the most similar frame for $v_i$, all frames in $u$ are considered by their similarity weights. In this way the transform $W$ becomes smooth and more resilient to noise frames.

**Clustering linear transforms.** Again we utilize clustering algorithm to extract all the possible transform patterns that map partial sequences to full sequences. Since the original number of transforms for $W$ obtained from randomly sampled trimmed and full pairs may be too large, with clustering we avoid the inefficient sampling of $W$. Meanwhile, during the clustering, some important transform patterns, e.g. mirroring, can stand out as they may originally only account for a small percentage in the whole pool of transforms.

In details, we first randomly generate training pairs, i.e. partial motion sequences and full motion sequences $\{(u_k, v_k)\}$ from $\mathcal{S}$. We then use Eq. (3) to get similarity matrix $M^{(k)}$ for motion $u_k$ and $v_k$. Instead of clustering directly on $W$, we perform clustering on all the similarity matrices $\{M^{(k)}\}$ to get representative cluster centers $\{M_i\}$. We normalize each row for $M_i$, and then calculate the index according to Eq. (4) and obtain the corresponding transform matrix $W_i$. Again, during clustering, each $M^{(k)} \in \mathbb{R}^{T \times T}$ is flattened to 1d vector and we use $L_2$ norm as the distance measure.

Once we obtain a set of $\{W_i\}$, we can randomly sample $W$ from $\{W_i\}$ and apply it to $x$. So the linear transform can be formally represented as $\mathcal{F}_L(x) = Wx, W \sim \{W_i | W_i \in \mathbb{R}^{T \times T}\}$.

### 3.5 Training

We follow the standard pipeline for training a recognition model. For input $x$ with its action label $y$, we generate augmentations $x'$ (refer to Algorithm 1 in Appendix A1), and train the model with the loss function:

$$\mathcal{L}_{\text{total}} = \mathcal{L}(\mathcal{G}(x), y) + \mathcal{L}(\mathcal{G}(x'), y), \tag{5}$$

where $\mathcal{L}$ is the standard cross-entropy loss. In practice, in a batch, we augment the training samples with a ratio of $m_{\text{aug}}$, and the rest of the samples remains unchanged.

## 4 Experiments

### 4.1 Datasets and Settings

**Datasets.** We use four large-scale datasets, i.e, NTU60-RGBD [41], PKU-MMD [25], ETRI-Activity3D [15] and Kinetics [5]. The first three datasets are captured in indoor laboratory and household environments while the last one is collected from online videos.

**A new cross-dataset setting.** We construct a new multi-domain cross-dataset setting, mainly comprising actions in indoor environment. We gather shared 18 actions (see Appendix A2) using the first three datasets described above and each dataset is treated as one domain. We train on one domain and test on the rest two domains. Specifically, we define four domain transfer sub-settings, i.e. $N \to E, N \to P, E_A \to N, E_A \to P$ (each dataset is denoted with its first letter). Especially we use the adult split $E_A$ for training because in this way we keep the number of training samples for NTU and ETRI relatively the same. We do not include $P$ for training since $P$ majorly contains long and complete actions due to its annotation and it is relatively easy to perform sampling directly. We also reserve a subset $\bar{P}$ with mutually exclusive labels with $P$ as a prior dataset to evaluate in an "Oracle" case when we have certain knowledge about how complete actions look like.

**Evaluation Metric.** We use the average accuracy of the above four cross-dataset sub-settings to measure the performance of single domain generalization. Especially we use balanced accuracy [32, 39] to eliminate the influence of class bias. We train the model with ten different random seeds and report the mean accuracy.

For fair comparison with ST-Cubism [46], we also include $P51 \leftrightarrow N51$ with 51 action classes paired between NTU and PKU dataset, and $N12 \to K12$ with 12 action classes paired between NTU and Kinetics. We also follow their evaluation protocols when reporting results.

### 4.2 Implementation Details

**Data preparation.** We resize motion sequences from different domains to a fixed length $T = 64$. Following [42], we remove camera rotation and trajectory movement in the pre-processing stage and we further apply random 3D rotation as spatial augmentation for all methods.

Table 1: Comparison with other methods in our cross-dataset settings. The best result is in bold and the second best is with underlines.

| Method | $N \to E$ | $N \to P$ | $E_A \to N$ | $E_A \to P$ | Avg. |
|---|---|---|---|---|---|
| ERM | 54.9 | 70.5 | 42.4 | 49.7 | 54.4 |
| CCSA [35] | 56.0 | 72.2 | 43.6 | 51.7 | 55.9 |
| ADA [52] | 55.2 | 69.2 | 43.8 | 50.7 | 54.7 |
| ST-Cubism [46] | **59.1** | 71.7 | 45.4 | 52.4 | 57.1 |
| Skeleton-MAE [28] | 56.1 | 72.7 | 44.5 | 52.4 | 56.4 |
| HICLR [62] | 54.0 | 70.6 | 46.7 | 53.8 | 56.3 |
| Uniform sampling [8] | 56.7 | 69.4 | 45.3 | 51.3 | 55.7 |
| Mixup [61] | 55.0 | 69.9 | 44.5 | 52.2 | 55.4 |
| CropPad [23] | 56.8 | 70.1 | 44.0 | 50.9 | 55.4 |
| CropResize [47] | 57.5 | 70.3 | 45.5 | 50.6 | 56.0 |
| TSN [54] | 54.3 | 68.6 | 43.0 | 50.0 | 54.0 |
| Multiple-crop testing | 54.4 | **76.5** | 40.8 | 52.6 | 56.1 |
| OTAM+kNN [3] | 54.2 | 72.7 | 42.9 | 50.8 | 55.2 |
| Ours | 58.4 | 75.8 | **48.4** | **57.8** | **60.1** |

Table 2: Comparison with other methods in NTU → PKU setting with 51 actions.

| Method | $N51 \to P51$ |
|---|---|
| ERM | 66.3 |
| CCSA [35] | 67.3 |
| ST-Cubism [46] | 70.5 |
| Skeleton-MAE [28] | 70.3 |
| HICLR [62] | 66.5 |
| CropPad [23] | 69.0 |
| CropResize [47] | 67.0 |
| Ours | **72.2** |

Table 3: Comparison with ST-Cubism in multiple cross-dataset settings with HCN backbone.

| Method | Backbone | $N51 \to P51$ | $P51 \to N51$ |
|---|---|---|---|
| ERM | HCN | 57.6 | 50.5 |
| ST-Cubism (Tem) [46] | HCN | 60.0 | 52.7 |
| ST-Cubism (Spa) [46] | HCN | 59.6 | 50.8 |
| ST-Cubism [46] | HCN | 61.3 | **53.8** |
| Ours | HCN | **62.8** | 53.3 |

| Method | Backbone | $N12 \to K12$ |
|---|---|---|
| ERM | HCN | 14.4 |
| ST-Cubism [46] | HCN | 15.6 |
| Ours | HCN | **15.9** |

**Backbone models.** For our backbone model, we adopt a slightly modified version [26] of Adaptive GCN (AGCN) [42] with fewer blocks and only use the joint stream for speed and simplicity. We reduce the number of blocks from 10 to 4. The output channels for each block are 64, 64, 128 and 256. The kernel strides for each block are 1, 1, 2 and 2. We find that such a design improves the base generalizability partially because it better fits the reduced length of motion sequences and avoids overfitting. Besides our AGCN backbone, we also test on ST-GCN [57] and CTR-GCN [6], two representative GCN backbones. ST-GCN is a simple GCN that aggregates spatiotemporal information without elaborate design. CTR-GCN adopts a multi-level feature design and has much more parameters than our AGCN. HCN is a two-stream convolutional network used by [46]. We use it for fair comparison in $P51 \leftrightarrow N51$ and $N12 \to K12$ settings.

**Training details.** We set training hyper-parameters the same as [42]. Furthermore, we set $\lambda_T = 0.1$, the number of linear transforms $N_{tr} = 20$, the number of background poses $N_{bkg} = 10$, and $m_{aug} = 0.75$. We set $\alpha = 0.1$ for sampling $t_p$. We fix the resampling method by randomly sampling a segment with length ratio $r$ between 0.7 and 1.0. We use $k$-means for clustering boundary poses and linear transforms. Parameter settings for $P51 \leftrightarrow N51$ and $N12 \to K12$ are provided in Appendix A3.

### 4.3 Results

**Baselines.** Besides ST-Cubism [46] which is by far the only method that reports result for cross-dataset settings, we compare with a wide range of baseline methods. **(1) General domain generalization approaches:** latent feature alignment CCSA [35] and adversarial augmentation ADA [51, 52]. **(2) Self-supervised learning approaches:** ST-Cubism [46] that solves jigsaw puzzles in spatial (Spa) and temporal (Tem) dimensions, Skeleton-MAE (partially borrowed from [28]) that

Table 4: Effect of each component and the effect of using different prior datasets for our method. All ablation experiments in this table have the resampling step.

| Method | $N \to E$ | $N \to P$ | $E_A \to N$ | $E_A \to P$ | Avg. |
|---|---|---|---|---|---|
| ERM | 54.9 | 70.5 | 42.4 | 49.7 | 54.4 |
| CropPad [23] | 58.0 | 67.9 | 45.2 | 51.7 | 55.7 |
| Nonlinear $\mathcal{F}_N$ | 57.7 | 70.8 | 48.3 | 54.5 | 57.8 |
| Linear $\mathcal{F}_L$ | 58.6 | 72.8 | 45.3 | 52.2 | 57.2 |
| Linear(Self)+Nonlinear(Self) | 58.4 | 75.8 | 48.4 | 57.8 | 60.1 |
| Linear($\bar{P}$) +Nonlinear(Self) | 58.2 | 76.2 | 48.5 | 57.7 | 60.2 |
| Linear(Self)+Nonlinear($\bar{P}$) | **58.7** | **76.4** | 48.4 | 58.4 | **60.5** |
| Linear($\bar{P}$) +Nonlinear($\bar{P}$) | 57.9 | 76.3 | **49.2** | **58.6** | **60.5** |

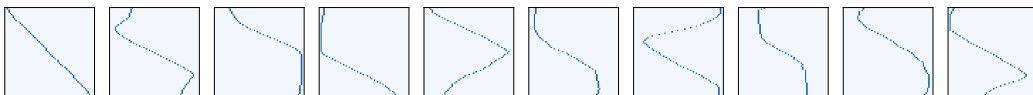

Figure 3: Visualization for selected linear transform matrices $\{W_i\}$ via clustering using training sets $N$ and $E_A$.

learns skeleton representation in the fashion of masked auto-encoder, and HICLR [62] which is a representative pre-training method via contrastive learning. **(3) Augmentation-based approaches:** mixed sample data augmentation Mixup [61], Crop and reflective padding (CropPad) [23], crop and resizing (CropResize) [47] and uniform sampling [8]. **(4) Alignment-based approaches:** OTAM [3] +kNN which uses nearest neighbour classifier [53] with a variant OTAM [3] as the dynamic time warping distance. **(5) Aggregation-based approaches:** Temporal segment network design [54] and multiple-crop testing (5-crop) which samples clips from a test sequence and average the output scores. Moreover, Empirical Risk Minimization (**ERM**) uses standard cross-entropy loss and serves as a performance lower bound. More implementation details are provided in Appendix A3.

The result is shown in Table 1. Our method, by explicitly exploring the action prior, improves ERM by 5.7% and outperforms the second best (ST-Cubism) by 3.0% in terms of average accuracy. All the self-supervised and augmentation-based methods improve the domain generalizability while the former perform better than the latter. However, neither of them can significantly improve the generalizability to deal with unseen data with large gaps. By learning transforms from training data, we are more flexible than handcrafted augmentations and perform especially better in recognizing long sequences ($P$ domain). Test-time aggregation only improves on one specific setting and warping-based matching with kNN is not as competitive as directly learning a deep classifier when given sufficient and diverse training data (See more discussion in Appendix A6). While sharing the same spirit that temporal alignment is important, we provide an alternative solution which combines raw training data and augmented "aligned" data from recovering and resampling complete actions.

We further transfer boundary poses and linear transforms learned from the 18-class subset to $N51 \leftrightarrow P51$ settings. In Table 3, we compare to ST-Cubism [46] following their protocol using HCN [21] backbone. For $N51 \to P51$, as in Table 2 and 3, our method outperforms ST-Cubism, showing the effectiveness when there are more action classes. For $P51 \to N51$, ours is better than the temporal part of ST-Cubism but slightly worse than the whole ST-Cubism, which can be explained that $P51$ generally already consists of complete actions. For the challenging 2D setting $N12 \to K12$, we use linear transform module only and also obtain slightly better performance than ST-Cubism.

### 4.4 Analysis and Discussion

**Effect of each component.** We examine the effect of the non-linear transform $\mathcal{F}_N$, i.e. the boundary conditioned extrapolation, and the linear transform $\mathcal{F}_L$. As shown in Table 4, both $\mathcal{F}_N$ and $\mathcal{F}_L$ improve the ERM baseline in terms of the average accuracy, showing they are crucial to construct complete actions and validating our two-step design. As a comparison, we see that recovering with handcrafted CropPad only leads to limited improvement (1.3%). Moreover, in cases when the action completeness is less significant in the target domain (e.g. $N \to E$), $\mathcal{F}_L$ itself may be sufficient.

Table 5: Effect of linear transforms.

| $N_{tr}$ | $N{\to}E$ | $N{\to}P$ | $E_A{\to}N$ | $E_A{\to}P$ | Avg. |
|---|---|---|---|---|---|
| ERM | 54.9 | 70.5 | 42.4 | 49.7 | 54.4 |
| 3 | 57.8 | 69.8 | **49.2** | **58.7** | 58.9 |
| 5 | 58.2 | 72.4 | 48.9 | 57.2 | 59.2 |
| 10 | 58.2 | 73.7 | 48.2 | 56.9 | 59.3 |
| 20 | **58.4** | **75.8** | 48.4 | 57.8 | **60.1** |

Table 6: Effect of background poses.

| $N_{bkg}$ | $N{\to}E$ | $N{\to}P$ | $E_A{\to}N$ | $E_A{\to}P$ | Avg. |
|---|---|---|---|---|---|
| ERM | 54.9 | 70.5 | 42.4 | 49.7 | 54.4 |
| None | **58.6** | 73.7 | 47.6 | 55.3 | 58.8 |
| 5 | 58.1 | **76.3** | 48.1 | 57.2 | 59.9 |
| 10 | 58.4 | 75.8 | 48.4 | **57.8** | **60.1** |
| 20 | 58.4 | 75.7 | **48.5** | 57.4 | 60.0 |

Table 7: Comparison with other learning variants in our method.

| Method | $N{\to}E$ | $N{\to}P$ | $E_A{\to}N$ | $E_A{\to}P$ | Avg. |
|---|---|---|---|---|---|
| ERM | 54.9 | 70.5 | 42.4 | 49.7 | 54.4 |
| $\mathcal{F}_{NN,extrap}(\cdot)$ | **58.6** | 74.3 | 46.3 | 56.1 | 58.8 |
| $\mathcal{F}_{NN,infill}(\cdot)$ | **58.6** | 73.8 | 46.7 | 55.6 | 58.7 |
| Ours | 58.4 | **75.8** | **48.4** | **57.8** | **60.1** |

Table 8: Per-class accuracy improvement of our proposed method compared to ERM.

| Action | Avg. acc. across four settings |
|---|---|
| Phone calling | 38.8(**+25.3**) |
| Hand waving | 71.3(**+22.1**) |
| Clapping | 54.7(**+15.0**) |
| Pointing finger | 92.5(**+9.7**) |
| Taking off clothes | 88.0(**+3.1**) |

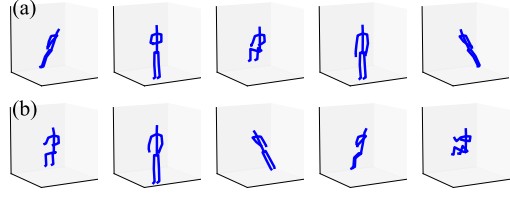

Figure 4: Visualization for the boundary pose clustering result $\{p_i\}$ when $N_{bkg} = 5$. (a) Pose clusters for training set $N$ and (b) Pose clusters for training set $E_A$.

Figure 5: Examples of some recovered complete actions. The skeletons in **blue** are raw inputs and the skeletons in **sky blue** are new frames generated by our method.

Since $\mathcal{F}_L$ and $\mathcal{F}_N$ learned from different datasets are transferable, we can also use $\bar{P}$ as a high quality prior dataset to replace the original training data (denoted as Self in Table 4) to learn $\mathcal{F}_L$ and $\mathcal{F}_N$. In this way we can investigate the performance upper bound of our method. By utilizing $\bar{P}$ we can further improve the average accuracy from 60.1 to 60.5. This shows that it would be more beneficial to directly learn from nearly complete actions since it is our aim to reconstruct complete actions from partial observations. On the other hand, the marginal improvement of 0.4% also indicates that even from the training data we already learn transforms and boundary poses in a quite decent manner.

**Parameter analysis.** We examine three important parameters: number of clustered boundary poses $N_{bkg}$, number of clustered linear transforms $N_{tr}$, and context-aware coefficient $\lambda_T$. The results are in Table 5 and 6. As for $N_{tr}$, a too small number of clusters will hurt the overall performance by around 1.2%, which shows that too few linear transforms cannot fully represent the various global motion patterns. As for $N_{bkg}$, we empirically find that the number is not a very important issue probably because the variety of boundary poses does not significantly affect the semantics. However, directly extending with the raw first frame (shown as "None") will hurt the performance by 1.3%, indicating the importance of boundary pose clustering. We also investigate the parameter sensitivity of $\lambda_T$. We find that $\lambda_T = 0.1$ works well and generally it is not very sensitive (see result in Appendix A4).

**What is learned for augmentation?** In order to better understand what is learned in $\mathcal{F}_L$ and $\mathcal{F}_N$, we visualize the clustering results. Fig. 4 visualizes the clustered background poses. We find that the background poses mainly include common rest poses such as standing and sitting with different rotations. These poses do not always appear in all action sequences, so clustering from the whole dataset is reasonable. Fig. 3 visualizes the clustered linear transform matrices. We find that $\mathcal{F}_L$ mainly learns temporal shifting and reflection operation from the training data, which indicates the presence of approximate symmetric pattern of many complete actions (see matrices in ">" and "<" shapes). We can also interestingly conclude that the CropPad transform [23] actually fits our complete action prior well as it appears in the set of $\mathcal{F}_L$ transforms (such as the second matrix in Fig. 3). Ours

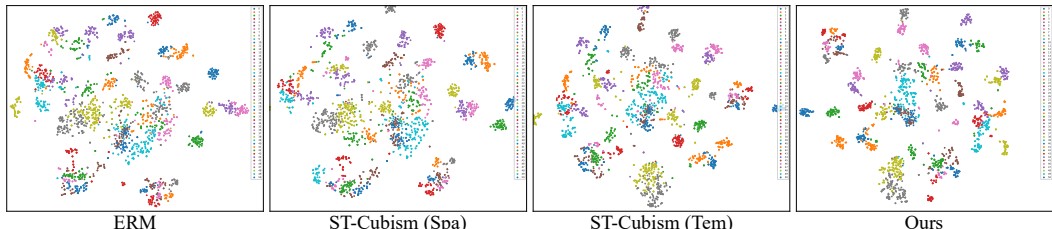

| ERM | ST-Cubism (Spa) | ST-Cubism (Tem) | Ours |

Figure 6: The t-SNE [49] visualization of the feature embedding for the test set in $N51 \rightarrow P51$ setting with HCN backbone. The color indicates the groundtruth label. Our method learns a more discriminative feature embedding.

Table 9: Performance of our method on different backbones.

| Backbone | $N \rightarrow E$ | $N \rightarrow P$ | $E_A \rightarrow N$ | $E_A \rightarrow P$ | Avg. |
|---|---|---|---|---|---|
| ST-GCN [57] | 51.7 | 67.7 | 37.8 | 44.5 | 50.4 |
| ST-GCN+Ours | 56.7 | 69.7 | 46.1 | 49.0 | 55.4 |
| CTR-GCN [6] | 54.7 | 72.9 | 46.6 | 50.9 | 56.3 |
| CTR-GCN+Ours | 63.6 | 81.4 | 53.9 | 58.3 | 64.3 |

is more flexible as we can learn a set of possible transforms. Fig. 5 provides examples of recovered complete actions from partial observations. In Fig. 6, we show that with our augmentation we learn a more discriminative feature embedding compared to some other baselines.

**Comparison with other learning variants.** Note that it is also possible to perform infilling and extrapolation for skeleton sequences by training an action completion network. In Table 7 we investigate the performance when learning by neural networks. We compare to the design of learning to extrapolate without conditioning on boundary poses, as well as the design of learning to infill instead of performing linear interpolation. For both variants we adopt the network design in [28] and more details are provided in Appendix A3. We find that in terms of average accuracy, both learning to extrapolate ($\mathcal{F}_{\text{NN,extrap}}$) and learning to infill ($\mathcal{F}_{\text{NN,infill}}$) are not as effective as our proposed learning-free interpolation, which shows that (1) assigning boundary pose via clustering is necessary as the network itself is difficult to learn automatically; (2) infilling by linear extrapolation is actually a simple but effective design as the infilling quality is good and robust for domain generalization.

**Per-class results.** Table 8 shows the performance improvement for selected action categories. Our method can significantly improve the accuracy on several hand-related actions such as *phone calling*, *hand waving* and *clapping*. In the case of temporal mismatch, these actions are easily confused with other similar actions if only trained from partial observations. If an action can be well recognized by partial segments, the improvement may be small (e.g. *taking off clothes*).

**Generalizability for different backbones.** We further apply our method on ST-GCN [57] and CTR-GCN [6] to see whether various backbones can benefit from the observed complete action prior. Note that CTR-GCN extracts temporal feature in a multi-scale fashion, which is more advanced than AGCN [42]. We observe consistent improvement over ERM, from 50.4 to 55.4 for ST-GCN and from 56.3 to 64.3 for CTR-GCN, as is shown in Table 9. This shows that the domain gap caused by temporal mismatch is difficult to be mitigated by only designing network architecture itself.

## 5 Conclusion

In this paper we present a recover-and-resample augmentation approach to deal with the single domain generalization problem for skeleton action recognition. By exploring the complete action prior, we recover complete actions with learned boundary poses and global linear transforms via clustering. Experiments on a cross-domain setting with three datasets validate our framework. In the future, we plan to investigate more effective resampling approaches, e.g. positional encoding, to further incorporate temporal resampling information into our whole framework.

## Acknowledgements

We thank all the reviewers for their useful suggestions. This work was supported by the National Natural Science Foundation of China (62220106003), the Research Grant of Beijing Higher Institution Engineering Research Center, the Tsinghua-Tencent Joint Laboratory for Internet Innovation Technology, and the Tsinghua University Initiative Scientific Research Program.

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

# 6 Appendix

## A1. Algorithm

The algorithmic description of our recover-and-resample augmentation framework is provided in Algorithm 1.

---
**Algorithm 1** Recover and Resample

---
**input:** a sample $x \in \mathbb{R}^{T \times J \times 3}$ with length $T$ and joint number $J$, dataset $\mathcal{S}$, resampling function $W_{\text{resample}}$, coefficient $\alpha$.
**output:** an augmented sample $x' \in \mathbb{R}^{T \times J \times 3}$.
  Cluster background poses $\{p_i\}$ from $\mathcal{S}$. (Sec. 3.3)
  Cluster linear transforms $\{W_i\}$ from $\mathcal{S}$. (Sec. 3.4)
  Define a motion infiller $\mathcal{F}(\cdot)$.
  Assign $p'$ as the boundary pose for $x$ from $\{p_i\}$ using Eq. (2).
  Sample $t_p \sim \beta(\alpha, \alpha)T/2$ and perform extrapolation $x = \mathcal{F}(x, p', t_p)$.
  Sample $W$ from $\{W_i\}$, apply $x = Wx$.
  $x' = W_{\text{resample}}(x)$.

---

## A2. Description on Datasets and Evaluation Settings

**Cross-dataset settings.** Here we provide more details for our cross-dataset settings. The shared 18 action classes for NTU60-RGBD [41], PKU-MMD [25] and ETRI-Activity3D [15] are listed in Table 10. We evaluate on four cross-dataset sub-settings, i.e. $N \to E$, $N \to P$, $E_A \to N$ and $E_A \to P$. We use cross-subject protocol for dividing training and test splits, and the number of training and test samples for each domain is shown in Table 11. For each domain, we use *training* set for model training and *test* set for evaluation. For term of use, NTU60-RGBD [41] is free for research and non-commercial use. We submitted a license agreement to ETRI-Activity3D [15] website for downloading the dataset. License for PKU-MMD [25] is not stated on its official homepage.

Table 10: Action labels used in our cross-dataset settings.

| Action label | | | |
|---|---|---|---|
| eat | drink | brush teeth | brush hair |
| wear clothes | take off clothes | put on/take off glasses | read |
| write | phone call | play with phone | clap |
| bow | handshake | hug | hand wave |
| point finger | fall down | | |

Table 11: Number of training and test samples for each dataset split.

| Domain | # samples in training set | # samples in test set |
|---|---|---|
| $N$ | 12651 | 5212 |
| $E_A$ | 13568 | 7008 |
| $P$ | 6823 | 968 |
| $E$ | 26930 | 13350 |
| $\bar{P}$ | 12017 | 1736 |

**Evaluation details.** For skeleton action recognition, most works [42, 6] report best results on the test set since there is no official validation set. We also follow this evaluation protocol for our domain generalization task. We train and evaluate for each sub-setting separately, and use average accuracy for evaluating the overall performance. Moreover, we run each experiment ten times with different random seeds and report mean accuracy to reduce performance oscillation. As the training sets for $N$ and $E_A$ (the adult split) are approximately of the same sizes, we choose to use $E_A$ for training, which offers a fair basis and gives us a better understanding of the task difficulty for different sub-settings.

## A3. Additional Implementation Details

**Linear transform clustering.** In order to cluster a set of linear transforms, we construct a set of training pairs, i.e. partial action sequences and full action sequences $\{(u_k, v_k)\}$ from training data $\mathcal{S}$. Although it is possible to randomly sample starting and end points for a sequence $v$ to generate $u$, we propose to sample from some fixed interval timestamps for simplicity and regularity. Those timestamps are $0.0, 0.125, 0.25, 0.375, 0.5, 0.625, 0.75, 0.875, 1.0$. For each training sample $v$, we sample $u$ according to the below pairs of starting and end points.

$$[(0.0, 1.0), (0.0, 0.5), (0.5, 1.0), (0.25, 0.75), (0.125, 0.625), (0.375, 0.875), (0.0, 0.75), (0.25, 1.0)]$$

**Resampling stage.** We mainly follow the CropResize [47] operation for our resampling stage, which first samples sequence length from uniform distribution $U(r_1, r_2)$ and then randomly samples a temporal segment with this length. Since effective resampling approach is not the primary goal of our paper, we fix it in our framework.

**Training details.** For training AGCN, we follow [42] and use SGD optimizer with an initial learning rate of 0.1 and weight decay of 0.0001. We train the model for 50 epochs. The learning rate is reduced at 30th and 40th epoch with a decay of 0.1. For HCN we use the same training configuration as [46]. Parameter settings for other experiments are listed below. For $N12 \rightarrow K12$ and $N51 \leftrightarrow P51$, we set $N_{\text{tr}} = 10$. For $N51 \rightarrow P51$, $N_{\text{bkg}} = 10$ and for $P51 \rightarrow N51$, $N_{\text{bkg}} = 5$. The boundary poses and linear transforms learned from the corresponding 18-class subset are transferred to $N51 \leftrightarrow P51$ settings for efficiency. Also for $N12 \rightarrow K12$, we set $m_{aug}$ to a smaller value of 0.5 and omit $\mathcal{F}_N$ due to very high level of noise and missing joints in $K12$ dataset. We implement our method and other baselines using PyTorch [36] and conduct experiments on a single NVIDIA RTX 2080Ti. We will also release code in Jittor [13] implementation which supports faster training and inference.

**Design of learning variants for comparison.** As in Table 7, for learning to extrapolate ($\mathcal{F}_{\text{NN,extrap}}$), we train a completion network using masked and full sequences, and during augmentation, we squeeze the original sequence to half of its original size and let the network extrapolate on both sides. We also compare with the design of learning to infill ($\mathcal{F}_{\text{NN,infill}}$) instead of performing linear interpolation. Again we mask out a segment in the center of a motion sequence and use paired sequences to train an infilling network.

**Other baseline methods.** Here we provide brief descriptions and more implementation details for other baseline methods.

**(1) General domain generalization approaches.** CCSA [35] improves generalizability by aligning samples of the same label in the feature space via contrastive loss. For CCSA, the weight for contrastive loss is set to 0.1, and the margin for contrastive loss is set to 1.0. ADA [51] improves generalizability by augmenting with adversarial samples. ADA uses image-based perturbation in the original paper. We adapt it to the adversarial skeleton perturbation [52] as we find it yields better results. The training set is expanded with adversarial samples at 15th and 30th epochs. For adversarial augmentation we set the semantic regularizer to 1.0, learning rate to 1.0 and optimization steps to 5.

**(2) Self-supervised learning approaches.** ST-Cubism [46] improves generalizability by solving auxiliary jigsaw puzzles. The spatial stream (Spa) and temporal stream (Tem) solves jigsaw puzzles in spatial and temporal dimensions respectively and the final prediction ensembles scores from the two streams. For ST-Cubism, the loss weight for jigsaw puzzle term is set to 0.1. Skeleton-MAE improves generalizability by learning to reconstruct the input skeletal motion sequence. For Skeleton-MAE, we borrow the masked auto-encoder module and reconstruction loss from [28], and change the backbone to AGCN. During training, we randomly mask out a clip with a length ratio of 0.3 and let the model to reconstruct it. The weight for the reconstruction loss is set to 1.0. HICLR [62] is a skeleton representation learning method, which improves generalizability by pre-training a feature encoder via contrastive learning. For HICLR, we follow their supervised evaluation by first pre-training the encoder and then performing standard training using the training set. We adapt the backbone to AGCN as well.

**(3) Augmentation-based approaches.** Mixup [61] generates augmentations by linearly interpolating between a pair of samples as well as their labels. For Mixup, training samples are augmented with Mixup with a probability of $p = 0.5$. CropPad [23] randomly crops a segment from the motion sequence and performs reflective padding on both sides. For CropPad, we use the crop ratio $\gamma = 6$.

CropResize [47] randomly crops a segment from the motion sequence and resizes it to its original length. For CropResize, we randomly sample a segment with a length ratio between 0.7 and 1.0, which we find works the best. We also set the resampling process in our method the same as this baseline. For CropPad, CropResize, and spatial augmentations, we set $m_{\text{aug}} = 0.75$ which is the same as ours. Uniform sampling [8] resizes a sequence by first uniformly dividing the input sequence into segments and then randomly sampling one frame within each segment. Such a sampling method can make full use of all frames in the raw motion sequence and therefore generate strong augmentations.

**(4) Alignment-based approaches.** We use the feature encoder from the model trained with standard cross-entropy loss (ERM), and transform raw skeleton sequences into downsampled feature sequences (16 frames for each sample). We then use a kNN ($k = 5$) classifier [53] with warping-based distance OTAM [3] which is a representative warping-based distance measure used in few-shot video classification that deals with temporal mismatch. The final result comes from the ensemble of OTAM+kNN and the ERM model. We find that using distance in the raw joint space is inferior than in the feature space. Note that we implement temporal alignment method without further pre-training the feature encoder. Please refer to self-supervised learning for feature pre-training approaches.

**(5) Aggregation-based approaches.** We build TSN [54] upon the AGCN backbone. The input sequence is uniformly divided into 3 clips before feeding to the backbone and we aggregate their features using average pooling. We implement multiple-crop testing as a post-processing method. Given a test sequence we sample 5 clips with length ratios of 0.5 and 0.75 and average the prediction for these clips altogether with the original sequence. The starting and end point pairs of the sampled clips are $[0, 0.5], [0.25, 0.75], [0.5, 1.0], [0, 0.75], [0.25, 1.0]$ in our experiments.

## A4. Additional Experimental Results

**Detailed results for ablation study.** Table 12 presents results for the experiment of **parameter analysis** in the main paper. We find that $\lambda_T$ is generally not very sensitive, as choosing $\lambda_T$ over a large range (0.001 to 1) does not significantly affect the average accuracy. Table 13 provides detailed results for the per-class performance.

Table 12: Effect of coefficient $\lambda_T$ in context-aware linear transforms.

| $\lambda_T$ | $N \to E$ | $N \to P$ | $E_A \to N$ | $E_A \to P$ | Avg. |
|---|---|---|---|---|---|
| ERM | 54.9 | 70.5 | 42.4 | 49.7 | 54.4 |
| 1 | 58.0 | 73.5 | 48.0 | 58.3 | 59.4 |
| 0.1 | **58.4** | **75.8** | 48.4 | 57.8 | **60.1** |
| 0.01 | 58.0 | 73.3 | **48.5** | **58.4** | 59.5 |
| 0.001 | 58.1 | 73.7 | 47.9 | 57.4 | 59.3 |

Table 13: Per-class results of our proposed method compared to ERM.

| Actions | $N \to E$ | $N \to P$ | $E_A \to N$ | $E_A \to P$ | Avg. |
|---|---|---|---|---|---|
| Phone calling | 20.8(+5.7) | 49.6(+49.1) | 32.7(+24.5) | 51.8(+22.0) | 38.8(**+25.3**) |
| Hand waving | 80.8(+21.5) | 96.8(+7.7) | 41.8(+21.1) | 65.9(+41.7) | 71.3(**+22.1**) |
| Clapping | 58.8(+23.1) | 90.0(+5.4) | 26.9(+14.4) | 43.2(+17.2) | 54.7(**+15.0**) |
| Pointing finger | 94.0(+2.7) | 100.0(+0.0) | 77.6(+22.9) | 98.2(+13.1) | 92.5(**+9.7**) |
| Taking off clothes | 98.5(+1.3) | 93.1(+2.9) | 70.9(+5.3) | 89.6(+2.7) | 88.0(**+3.1**) |

**Feature representations.** We measure the similarity of two skeletons using $L_2$ norm in the joint coordinate space. However, it is also possible that we measure it in the latent space. Here we investigate the performance difference between these two feature representations. Specifically, we train a skeleton auto-encoder with MLP using training data and use it to obtain latent feature for the skeleton in each frame. For $\mathcal{F}_N$ we perform boundary pose clustering in the latent feature space, and for $\mathcal{F}_L$ we calculate similarity matrices using latent feature. As in Table 14, we observe a lower but very close performance if using latent feature. This shows that our method does not depend on specific skeleton representations.

Table 14: Comparison between different skeleton representations for our method.

| Representation | $N \rightarrow E$ | $N \rightarrow P$ | $E_A \rightarrow N$ | $E_A \rightarrow P$ | Avg. |
|---|---|---|---|---|---|
| ERM | 54.9 | 70.5 | 42.4 | 49.7 | 54.4 |
| Ours (joint space) | **58.4** | **75.8** | **48.4** | **57.8** | **60.1** |
| Ours (latent space) | **58.4** | 74.9 | 48.3 | 57.6 | 59.8 |

Table 15: Comparison between different clustering algorithms.

| Method | $N{\rightarrow}E$ | $N{\rightarrow}P$ | $E_A{\rightarrow}N$ | $E_A{\rightarrow}P$ | Avg. |
|---|---|---|---|---|---|
| Ours ($k$-means) | 58.4 | 75.8 | 48.4 | 57.8 | 60.1 |
| Ours (agglomerative) | 58.5 | 74.7 | 48.0 | 56.7 | 59.5 |

Table 16: Comparison between different resizing strategies and segment lengths in the resampling stage. $r$ denotes the range of length for sampled segments.

| Resizing | range $r$ | $N{\rightarrow}E$ | $N{\rightarrow}P$ | $E_A{\rightarrow}N$ | $E_A{\rightarrow}P$ | Avg. |
|---|---|---|---|---|---|---|
| linear | [0.3,1.0] | 57.4 | 71.5 | 47.9 | 56.2 | 58.2 |
| linear | [0.5,1.0] | 58.4 | 74.7 | 48.1 | 56.3 | 59.4 |
| linear | [0.7,1.0] | 58.4 | 75.8 | 48.4 | 57.8 | 60.1 |
| random | [0.7,1.0] | 57.9 | 75.6 | 48.3 | 57.1 | 59.7 |

**Evaluation on different clustering algorithms.** We investigate whether our method is compatible with different clustering algorithms. We conduct experiments using agglomerative clustering [37], which is another type of clustering algorithm different from $k$-means. The result is in Table 15. We see that the average accuracy is lower but still close and outperforms other baseline methods, which shows that our method is generally not sensitive to the choice of clustering algorithm. In the experiment, we keep hyper-parameters (e.g. number of clusters) the same as $k$-means. The result may be further improved by tuning more suitable parameters.

**Evaluation on resampling methods.** For our experiments CropResize is used for the resampling stage. In Table 16, we investigate the length of segments $r$ as a key parameter. We find that sampling too short segments may introduce many noisy augmentations and hampers the overall performance. To resize the sampled segment to a fixed length sequence, linear and random frame sampling are also investigated. The results are shown to be generally comparable.

**Evaluation on standard in-domain action recognition.** Although the proposed method mainly deals with cross-dataset settings, we find it can also work as an augmentation strategy for standard in-domain skeleton action recognition with training and testing within the same dataset. In Table 17, we evaluate on the in-domain recognition on NTU-18 subset and ETRI Adult-18 subset from our constructed cross-dataset settings, as well as on the standard full NTU-60 dataset [41]. We adopt full AGCN backbone for N-18 and $E_A$-18, and use full 2s-AGCN [42] for NTU-60. We set a smaller weight for augmentation $m_{aug} = 0.25$. The performance improvement of our method is comparable to uniform sampling [8], which is considered as an effective temporal augmentation. Note that for cross-dataset settings we surpass uniform sampling by 4.4% (see Table 1) since uniform sampling only samples the observed sequence. On the other hand, the fact that our improvement on N-18 and $E_A$-18 is much smaller than in cross-dataset settings indicates that the training and test split of a dataset are generally well aligned. Our cross-dataset setting magnifies the issue of temporal mismatch and presents a more practical setting as in real-world applications.

Table 17: Result of our method for standard skeleton action recognition.

| Method | N-18 | $E_A$-18 | NTU-60 |
|---|---|---|---|
| ERM | 86.0 | 89.2 | 88.7 |
| Uniform Sampling [8] | 86.5 | **90.3** | 89.4 |
| Ours | **86.8** | 90.1 | **89.5** |

Table 18: Comparison between spatial and temporal augmentations.

| Method | $N \to E$ | $N \to P$ | $E_A \to N$ | $E_A \to P$ | Avg. |
|---|---|---|---|---|---|
| ERM | 54.9 | 70.5 | 42.4 | 49.7 | 54.4 |
| Shear [23] | 54.9 | 69.7 | 42.6 | 49.2 | 54.1 |
| Bone scaling [58] | 54.6 | 68.7 | 42.6 | 50.0 | 54.0 |
| Translation [11] | 55.2 | 68.8 | 43.3 | 50.3 | 54.4 |
| Ours (temporal) | **58.4** | **75.8** | **48.4** | **57.8** | **60.1** |

Table 19: Standard deviation of our cross-domain action recognition.

| Method | Avg. accuracy | Std |
|---|---|---|
| ERM | 54.4 | 0.8 |
| Ours | 60.1 | 0.6 |

**Spatial augmentation v.s. temporal augmentation.** In Table 18, we examine how spatial augmentation and temporal augmentation contribute to the improvement of generalizability, as they both prove to be effective in the self-supervised skeleton representation learning [23]. We choose typical spatial augmentations including shear transform [23], bone length scaling [58] and Gaussian noise addition on joints [11]. To compare with spatial augmentations, we set shear ratio [23] $\beta = 0.1$, and set the translational noise with a normal distribution $N(0, 0.01)$. For bone scaling, we randomly scale each bone with a length ratio between 0.9 and 1.1. We demonstrate that our method, in the form of temporal augmentation, is more effective than direct spatial augmentation. This shows that direct spatial transformation is more likely to change the semantics, especially for hand parts. Domain differences in human actions are easier to be handled in the temporal dimension than in the spatial dimension.

**More visualizations for $\mathcal{F}_L$.** In Fig. 8 we visualize the clustered linear transforms when $N_{tr} = 20$ and $N_{tr} = 5$ using training set $N$. We find that a small number of $N_{tr}$ (e.g. $N_{tr} = 5$) will hurt the diversity of linear transforms. In such a case, it is difficult to extract some important temporal action patterns (e.g. temporal shift and symmetry) via clustering.

**Error bars.** The domain generalization task normally brings the issue of performance oscillation. In Table 19 we report 10-run standard deviation for the average accuracy in our cross-dataset settings. We show that our improvement is statistically significant.

**Qualitative results.** In Fig. 9 we provide some qualitative examples showing our improvement over the baseline (ERM). With our proposed augmentation by action completion, we enable more accurate action recognition for long and full sequences when observing from partial action sequences.

## A5. Additional Analysis on Complete Action Prior

**Experimental details for discovering complete action prior.** Each input skeleton is flattened to a vector with a dimension of 75 (joint number $J = 25$). We design a simple skeleton auto-encoder consisting of six fully-connected layers and ReLU activation. The output dimensions for the fully-connected layers are 128, 64, 32, 64, 128, 75, respectively. The $L_2$ norm loss is adopted to train the auto-encoder in order to reconstruct the original input skeleton. During training, we sample frames from processed skeletal data (uniformly resized to 64 frames and the first frame is aligned to coordinate) as training data.

We train the auto-encoder separately for each dataset (PKU-MMD [25], NTU60-RGBD [41] and ETRI-Activity3D [15]) using data in our 18-class cross-dataset settings. We then use the trained model to transform all skeleton frames into latent features (extracted from the third layer with a dimension of 32). The feature diversity of dataset $\mathcal{S}$ at a certain timestamp $t$ is the standard deviation of the obtained latent features $z(\cdot)$ averaged over all feature dimensions $c \in \mathcal{C}$ and all skeleton

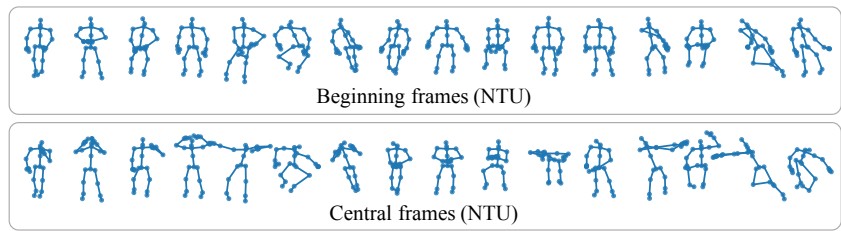

Figure 7: Visualization of beginning and central frames that are sampled from NTU split in our 18-class cross-dataset setting. The beginning frames which are usually rest poses have less diversity, while the central frames which contain rich action semantics usually have more diversity.

sequence samples $x \in \mathcal{S}$.

$$Diversity(\mathcal{S}, t) = \sqrt{\frac{1}{N_{\mathcal{S}} N_{\mathcal{C}}} \sum_{x \in \mathcal{S}} \sum_{c \in \mathcal{C}} (z(x_t)_c - \bar{z}(x_t)_c)^2} \qquad (6)$$

where $\bar{z}(x_t)$ is the mean of all latent features $z(x_t)$ for $x \in \mathcal{S}$, $N_{\mathcal{S}}$ is the size of dataset $\mathcal{S}$ and $N_{\mathcal{C}}$ is the number of latent feature dimensions.

**More analysis on the statistical pattern.** We provide more analysis for curve patterns in Fig. 1 (b). For PKU and NTU, as they are captured in labs and are generally clean with less occlusion and noise, they are of relatively low diversity and exhibit stronger level of complete action prior. For ETRI, as it is captured in home environment and is more challenging with larger noise, the frame-wise diversity would be higher. And as it is captured in real life with action boundaries that are hard to determine when collecting the dataset, the complete action prior is not as significant as NTU and PKU dataset. Fig. 7 shows examples of beginning frames and central frames for the NTU split in our cross-dataset settings, which is consistent with the statistical finding in Fig. 1 (b).

Also, we see that the global pattern differs across different datasets, for example, in the starting and end point diversity. For PKU and ETRI, they exhibit an overall temporal pattern of being symmetric. In contrast, the NTU dataset does not have such a symmetric pattern. There is also a shift for the peak diversity across datasets. The above differences are majorly caused by different definition of when an action starts and ends in different datasets, as can also be visualized in the Fig. 1 (a) example. These findings suggest that the domain gap between datasets, represented as temporal mismatch, may involve temporal shift, scaling and even symmetry/non-symmetry properties arising from the partial/full bias of human actions.

Note that the two observations presented in the main paper that many human actions are partial observations and that human actions generally exhibit completeness within a large group of samples are not contradictory. This can be best explained by that (1) within a large dataset, some action categories or samples are nearly complete while some other action categories or samples are partial segments of complete actions, and (2) even if the actions are not exactly complete, e.g. only comprising central segments, many of them obey a similar statistical pattern which is only less significant. Here the "completeness" is the opposite of having uniform diversity over time. Also keep in mind that the curves in Fig. 1 (b) only reflect the statistical pattern for a group of actions in a dataset. For example, although the curve for NTU seems non-symmetric, we can still have approximate symmetry operations in the clustered linear transforms (see Fig. 8). This can be best explained by that there exist action samples/categories that are symmetric in the NTU split, but they only account for a small portion.

**Other implications on the proposed method.** (1) The complete action prior generally relies on the assumption of clean skeleton data and daily routine pattern of human activities. Relying on this prior, our method is unable to work well when the source data is very noisy and contains very long motions with multiple stages, for example, taking $K12$ as source data. We find that $K12$ does not exhibit such strong prior as is observed in Fig. 1(b). (2) When the target domain does not show strong action completeness or contains high level of noise, the extrapolation module $\mathcal{F}_N$ may not have a very strong effect. This is partially shown in $N{\rightarrow}E$ and $N12{\rightarrow}K12$ settings. This also explains our finding that using linear transform $\mathcal{F}_L$ is generally sufficient for $N12{\rightarrow}K12$.

## A6. Discussion

**Discussion for other baseline methods**. We discuss some possible reasons that the alignment-based method is not as competitive as ours in the cross-dataset settings: (1) Given sufficient training data, the nearest neighbour classifier only memorizes training samples compared to network classifier and this in turn hampers its generalizability. Contrastive pre-training for the feature encoder with DTW-based loss may lead to better performance, but at a cost of much more computation. (2) Direct matching via kNN cannot fully take advantage of various types of augmentations, such as random rotation. (3) A better design of dynamic time warping is needed since there is domain shift across training and test datasets for either raw joint features or latent features. In widely-studied few-shot scenarios [3, 53], the training and test data normally come from the same distribution. For aggregation-based method, clips sampled without action completion cannot well handle the partial/full bias. However, these methods and our proposed method are not contradictory and can be combined for further improvement.

**Stochastic action completion**. In this paper we actually propose a stochastic action completion method that can be used to augment the training data. This stochastic action completion is more suitable for augmentation than a deterministic one. Note that $\mathcal{F}_N$ has the probability to retain its original motion due to our sampling strategy for length of extrapolation $t_p$. $\mathcal{F}_L$ also learns identical mapping as is shown in the learned transform matrices (diagonal matrix). In this way, even for the input of a complete action, our approach is still able to output a reasonable complete action by sampling an identical mapping. A supporting evidence is that for $P51 \rightarrow N51$ ($P51$ consists of generally complete actions), our method still improves the baseline ERM and even outperforms the temporal module of ST-Cubism [46].

**Number of new parameters.** During model training, the newly added parameters are two matrices, i.e., the boundary pose clusters of shape $(N_{\text{bkg}}, J, 3)$ and the linear transform clusters of shape $(N_{\text{tr}}, T, T)$. Here $N_{\text{bkg}} = 10$, $J = 25$, $N_{\text{tr}} = 20$, $T = 64$. So the number of new parameters of these two matrices is $\sim 90000$, which is less than two FC layers (suppose each layer is nn.Linear(256,256)). For other baseline methods, universal domain generalization methods CCSA [35], ADA [51] and handcrafted augmentation methods such as uniform sampling [8], Mixup [61], CropPad [23] and CropResize [47] do not introduce new parameters. Self-supervised learning methods often have a new branch along with the original network for learning auxiliary tasks, therefore introducing new parameters (usually several FC layers). So generally, our method and those self-supervised methods have a similar magnitude of new parameters.

**Limitations.** The limitations arising from the complete action prior are discussed in Appendix A5. *Other implications on the proposed method*. Moreover, we note that direct resampling from a complete action sequence might introduce ambiguity for some actions. For example, *standing up* and *sitting down* are two different actions, but both of them can be obtained by sampling from a complete action sequence in which a human stands up from sitting pose and then sits down. However, one should note that this problem also exists when constructing positive samples in contrastive self-supervised learning [11, 47], as resampling or reversing an action may change the semantics for some action categories. To deal with this problem, we plan to add positional encoding and temporal masks in the resampling stage to further encode temporal information about how the sequence is sampled. We leave it for future work.

**Broader societal impact.** Generally we don't find that our method has negative societal impacts. However, users should be careful about the result if action recognition along with our method is applied to critical human-related applications such as health monitoring and elderly fall detection. Current deep learning-based models cannot guarantee absolutely accurate action recognition.

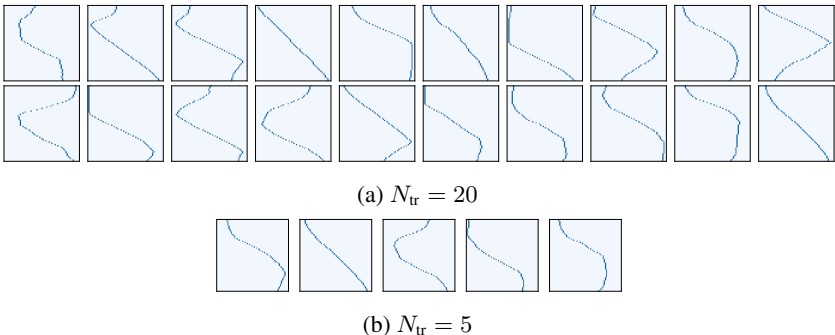

(a) $N_{\mathrm{tr}} = 20$

(b) $N_{\mathrm{tr}} = 5$

Figure 8: Visualization for the clustered linear transform matrices $\{W_i\}$ using training set $N$ when $N_{\mathrm{tr}} = 20$ and $N_{\mathrm{tr}} = 5$.

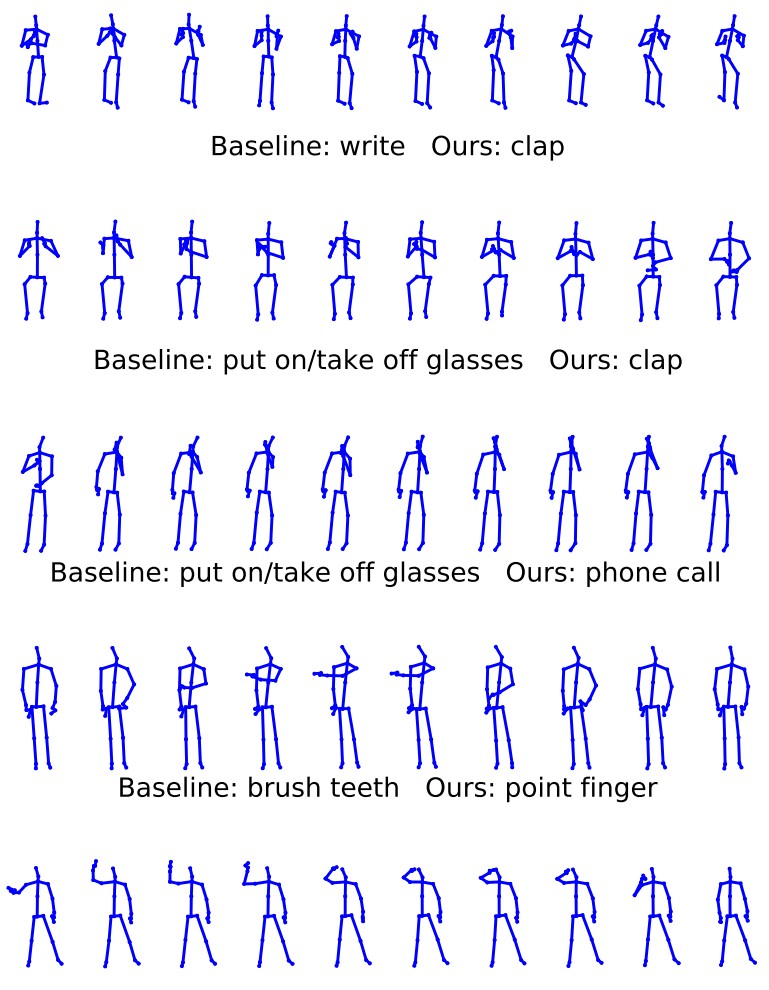

Baseline: write   Ours: clap

Baseline: put on/take off glasses   Ours: clap

Baseline: put on/take off glasses   Ours: phone call

Baseline: brush teeth   Ours: point finger

Baseline: brush hair   Ours: hand wave

Figure 9: Qualitative examples showing improvement of our method over the baseline (ERM).

