# OpenReview forum: "Recovering Complete Actions for Cross-dataset Skeleton Action Recognition"
_NeurIPS.cc/2024/Conference — NeurIPS 2024 poster_

### Official Review · Reviewer_UadT · 2024-07-09

**Soundness:** 3
**Presentation:** 3
**Contribution:** 4
**Rating:** 6
**Confidence:** 4

**Summary:**

This paper presents an innovative recover-and-resample augmentation framework to tackle the domain generalization challenge in skeleton-based action recognition. Utilizing the concept of a complete action prior, this method reconstructs entire action sequences from partial observations and resamples them to generate robust augmentations for unseen domains. The approach's effectiveness is confirmed through comprehensive experiments on three skeleton action datasets, showing considerable improvements over existing domain generalization methods.

**Strengths:**

1. Domain generalization of the skeleton based human action recognition is an interesting and unexplored research direction. The authors propose a good test bed for this task which can contribute to the society.

2. The authors propose a new augmentation method which is verified to be effective on various DG settings.

3. Comprehensive ablation studies are delivered by the authors to show the efficacy of the proposed approach.

**Weaknesses:**

1. Domain generalization is mostly handled via RGB image, since due to the background changes and the camera positions there are obvious large distribution shifts. Regarding the skeleton data, since they are mostly recorded in 3D coordinates without background information, where the domain gap has already been reduced. The authors should justify the need of domain generalization in the skeleton-based human action recognition field with more details in the introduction section. Since in the ultilized datasets, e.g., NTU120, it involves cross view and cross subject evaluation, which can also be regarded as domain generalization.

2.  In Section 4.3, how did the authors implement these approaches on the skeleton-based human action recognition methods? More details are expected to be enriched.


3. What is the number of parameters for the proposed approach? The authors are encouraged to make comparison between the proposed approach and the baselines regarding the number of parameters.

4. The authors should provide more clarification regarding why they chose those GCN backbones, e.g., HCN and CTRGCN. Maybe can be described as a small subsection in Sec. 4.

5. TSNE visualization is interesting on the test domain to see the learnt embeddings from different approaches.

**Questions:**

1. Can the authors elaborate more on why domain generalization is critical in the context of skeleton-based human action recognition, despite the inherent reduction in domain gaps provided by 3D coordinates?

2. Why did the authors choose HCN and CTRGCN as the GCN backbones for their approach? Can this be detailed further, perhaps as a subsection in Section 4?

3. Could the authors provide a more detailed explanation of how the approaches in Section 4.3 were implemented for skeleton-based human action recognition methods, including any specific techniques or modifications used?

4. What is the total number of parameters in the proposed approach, and how does it compare to the baseline methods?

5. Can the authors provide t-SNE visualizations of the learned embeddings from different approaches on the test domain to illustrate the effectiveness of the proposed method compared to others?

**Limitations:**

yes it is in appendix

---

> ### Author Rebuttal · Authors · 2024-08-07
>
> Dear Reviewer UadT:
>
> Thanks for your effort for reviewing our paper and giving kind suggestions. We really appreciate your positive comments on our paper.
>
> [Q1] Can the authors elaborate more on why domain generalization is critical in the context of skeleton-based human action recognition, despite the inherent reduction in domain gaps provided by 3D coordinates?
>
> [A1] Thanks for the suggestion. In the revision we will rewrite second paragraph in the introduction section elaborating on the importance of studying domain generalization in the context of skeleton-based human action recognition as following:
>
> Skeleton-based action representation has the advantage of removed background changes and camera positions, making it more robust compared to RGB representation. Despite that, generalizability under such a representation is still largely affected by the inherent spatiotemporal difference of 3D coordinates of a same action across domains. Essentially, cross-subject and cross-view settings are all cross-domain settings, yet they can be well addressed by designing more powerful backbones and applying geometric transformations, achieving high accuracy in the test set. However, we find that in the cross-dataset setting where source and target data come from different datasets, the performance degrades a lot (around or more than 20%) (comparing Table 1 and 15) and cannot be well remedied by the above approaches. This indicates drastic domain gaps in inherent feature of human actions across datasets, posing great challenges for real-life use and calling for research on domain generalization techniques for skeleton-based representation.
>
> Investigating action samples across multiple datasets, our observation is that a notable source of domain gap comes from the temporal mismatch of an action across different datasets (Fig. 1(a)), which is usually caused by different definition or cropping criterion of human motions…
>
>
> [Q2] Why did the authors choose HCN and CTRGCN as the GCN backbones for their approach? Can this be detailed further, perhaps as a subsection in Section 4?
>
> [A2] Thanks for the suggestion. We will add a new subsection to Section 4 briefly describing backbones used in the experiments, AGCN, HCN, CTR-GCN and ST-GCN. (1) We use Adaptive-GCN with reduced number of blocks for our main experiments for its good balance between efficiency and performance. (2) HCN is a convolutional network used by [41]. We use it for fair comparison in their setting as shown in Table 3. (3) ST-GCN and CTR-GCN are representative GCN backbones. CTR-GCN has much more parameters and multi-level feature design. ST-GCN is a simple GCN without elaborate design. We use them to show that designing network itself is hard to improve generalizability. (4) We will move the results of generalizability for different backbones to the main paper.
>
> [Q3] Could the authors provide a more detailed explanation of how the approaches in Section 4.3 were implemented for skeleton-based human action recognition methods, including any specific techniques or modifications used?
>
> [A3] Thanks for the suggestion. We will add more details to Appendix A3: other baseline methods.
>
> [Q4] What is the total number of parameters in the proposed approach, and how does it compare to the baseline methods?
>
> [A4] (1) During model training, the newly added parameters are only two matrices, i.e., boundary poses of shape (N_bkg, J, 3) and linear transforms of shape (N_tr, T, T). Here N_bkg=10, J=25, N_tr=20, T=64. Considering the number of parameters of AGCN is large, our learned parameters for action completion add very little to total parameters, (around ~90000), which are less than two FC layers (suppose each layer is nn.Linear(256,256)). (2) Comparing to other methods, general domain generalization methods CCSA, ADA and handcraft augmentation methods such as uniform sampling, mixup, CropPad and CropResize do not introduce new parameters. Self-supervised learning methods often have a new branch along with the original network for learning auxiliary tasks, therefore introducing new parameters (usually several FC layers). So generally, ours and those self-supervised methods have similar magnitude of new parameters.
>
> [Q5] Can the authors provide t-SNE visualizations of the learned embeddings from different approaches on the test domain to illustrate the effectiveness of the proposed method compared to others?
>
> [A5] Thanks for the suggestion, we will add t-sne plot in the revision, which is a good visualization tool for checking learned features.

---

> > ### Comment · Reviewer_UadT · 2024-08-09
> > **Response to the author**
> >
> > Thank you for the detailed response. I will keep my score as 6 according to the contribution of this work.

---

### Official Review · Reviewer_Vnjk · 2024-07-11

**Soundness:** 1
**Presentation:** 1
**Contribution:** 2
**Rating:** 5
**Confidence:** 3

**Summary:**

The work proposes a novel recover-and-resample augmentation framework for domain generalization with application to skeleton-based action recognition. The authors aim to tackle a specific issue when moving from one dataset to another, i.e. the temporal misalignment of actions of the same class. In the experimental analysis, they provide comparisons with existing approaches and ablation studies.

**Strengths:**

- The paper considers an important task, related to enriching the size, quality and variability of a dataset to finally improve the generalization abilities of trained models
- The methodology is modular, with intermediate outputs that can give valuable insights on the interpretation and the possibility of replacing the different tools with alternative choices
- The experimental analysis is very extensive. The protocol is well described and largely allows for reproducibility

**Weaknesses:**

- The introduction fails to guide the reader in understanding the motivations, methodologies and challenges. The motivations do not follow clear storytelling, and the description of the methodology is not fully comprehensible. Also, the language would need an improvement (the syntax of some sentences should be fixed)
- Also the SoA presentation fails to fully convince the reader of the value of the contributions of the proposed approach. The paper is related to different tasks and a more guided tour of the SoA would help to appreciate the contributions. Also, Domain generalization and Data Augmentation are two very important topics for the work, but the SoA discussion is very limited
- The methodology lacks technical details and appropriate, clear justifications for the different operations.
- It is not clear if the results are fully reproducible. To my understanding, the experimental protocol is not  used in other papers and there are no details on the implementations that have been adopted, so it is very difficult to judge the results (although the authors say the code will be available upon publications, indications on the adopted implementation would be useful)

**Questions:**

I would ask for clarifications on the following points

1.Introduction

- What do you mean by "It is actually a form that humans perform generally complete actions within large datasets"? If this is the case, what's the importance of alignment?
- "Although this prior can be detected by statistics in a general sense, in terms of individual samples, some exhibit strong action completeness while some are segments of their complete actions".
- On Fig. 1(b): the behaviour of NTU and PKU seems different from the one of ETRI (more variability in the initial poses for the latter), while in your comments ETRI is treated on par with the other two.
- "By studying the relationships between their raw and trimmed pairs, we can learn a temporal patterns inherited in human actions."
- You named part of the method "Complete action prior" but the term seems never used in the next sections
- In addition to these questions, a more general comment is that the introduction fails to convey the message, which should be a clear and convincing description of the methodology.  The picture does not fully help the understanding

2.Related Work

- Some sentences are not well justified, for instance, "These approaches partially improve the generalizability across datasets but are still bounded by the specific augmentation design." or "...but they do not make full use of the skeleton representations". An intuition of the meaning of statements of this type and in what sense the proposed approach is better would be beneficial

3.Method

- "On the other hand, the statistical finding that the initial poses of skeletal action sequences have low feature diversity (shown in Fig. 1 (b)) also validates our assumption: the boundary poses are to some extent constrained and similar to each other." This does not seem true for ETRI
- If in the training dataset the actions are all starting from the rest position (so they are all aligned) how the proposed method helps dealing with misalignment when changing the dataset?
- With the extrapolation, is the length T remaining the same?
- "Note that the above nonlinear transform is still unable to capture global and structural patterns inherited in human actions". Why?
- Why is it useful to reorganize the frames?
- What are the differences between the proposed work and [17]?
- Eq. 4: in my understanding k_i should be an index between 1 and T, but from the formula it does not seem so
- How do you generate the similarity matrix?
- Why should you cluster the W? The need for clustering is not clear
- From the loss function, it seems that each training sample is paired with an "augmented" sample, which is in contrast with the use of a parameter m_aug

4. Experiments

- A quantification (or at least an idea) of the level of misalignment of the different datasets would be useful for the interpretation of the results
- Reporting the results with P as training would still be useful as a sanity check
- The transfer sub-settings used in the experiments are different from [41] (which, according to the authors, is the only one reporting results for cross-dataset settings. Why this choice?
- What implementations have you used for obtaining the results in the tables?
- It is that Kinetic, mentioned in the adopted datasets, never appears in the experiments
- For the NTU datasets, have you considered the cross-view of the cross-subject problem?
- ERM corresponds to what version of the method?
- When reporting the per-class results, what about the actions that are not mentioned? It would be useful to have a more general idea
- The experiment briefly described in "Generalizability for different backbones" is actually important and should go in the main paper (discussed more in-depth)

Minor

- I would not mention the use of NNs for the motion infiller in the methodology description
- The description of the experiment with P51, N51, N12, and K12 should be moved above, with the other details on the transfer settings

[After reading the other reviews and the rebuttal from the authors, I'm happy to increase my score as most of my main concerns have been clarified]

**Limitations:**

Limitations and potential societal impacts are discussed in the Appendix

---

> ### Author Rebuttal · Authors · 2024-08-07
>
> Dear Reviewer Vnjk:
>
> Thanks for reviewing our paper. Due to space limit, we answer main questions regarding the soundness of the paper. Feel free to raise further questions.
>
> Introduction
>
> [Q1.1] What do you mean by "human performs generally complete actions within large datasets" and "in terms of individual samples, some exhibit strong action completeness while some are segments of their complete actions"?
>
> [A1.1] The two statements are not contradictory, which can be explained by within a large dataset, some action categories/samples are nearly complete and some action categories/samples are partial segments of complete actions. Meanwhile the overall statistics show the non-uniform diversity curve (meaning generally complete actions as opposed to uniform diversity). The insight is that the boundary poses and transforms mined from those complete categories/samples can be used to help perform action completion on those incomplete action ones, so we call them transferable knowledge (line47-48).
>
>
> [Q1.2] Clarify "the above nonlinear transform is still unable to capture global and structural patterns inherited in human actions"
>
> [A1.2] Sorry capture should be restore. Take phone calling as an example. Normally, a complete action phone calling consists of raising the arm, keeping hands close to head, and putting down the arm. During recovering, for a NTU sample (Fig 1(a)-NTU), we need to do mirroring so that the action becomes symmetric in time and thus becomes complete. For an ETRI sample (Fig 1(a)-ETRI), we have to first extrapolate the beginning pose to a rest pose, and then do mirroring so that the action ends with a rest pose as well. From above we see extrapolation only is not enough, and we still need global transforms to restore important properties of common complete actions, e.g., symmetry. Nonlinear transform refers to extrapolation here.
>
> [Q1.3] "By studying the relationships between their raw and trimmed pairs, we can learn temporal patterns inherited in human actions." Why reorganizing the frames?
>
> [A1.3] We use a segment of an action (i.e., trimmed sample) to reconstruct the original action (i.e, raw sample) with a linear transform (See Fig.2 u and v). It is an approximation, but in this way we can extract global patterns of human actions. For example, by sampling the first half segment of a symmetric action (e.g. Fig.1(a)-PKU phone calling) and trying to reconstruct the full, we can obtain a transform that functions like a mirroring operation. Note that the mirroring operation is a linear transform which can be characterized by a matrix of shape (T, T), and applying this matrix essentially means re-organizing existing frames since it does not introduce new frames. The learned linear transforms are not restricted to mirroring operation shown in this example, and can be shifting, scaling and mirroring, etc (see Fig.3).
>
> [Q1.4] If in the training dataset the actions are all starting from the rest position (so they are all aligned), … ?
>
> [A1.4] We have both recovering and resampling. During resampling, we randomly sample a segment from the recovered complete sequence, so the starting point of samples in the augmented training data is random and not necessarily the rest pose.
>
>
> [Q1.5] On Fig. 1(b): the behaviour of ETRI.
>
> [A1.5] Please refer to appendix A.5 (line629-632). NTU and PKU are captured in lab, while ETRI is captured at home with more noise and pose diversity. Yet clustering rest poses from beginning frames is still the best solution because they have least diversity.
>
>
>
> 3.Method
>
> [Q3.1] With extrapolation, is the length T remaining the same?
>
> [A3.1] Yes. See line 146-147. We squeeze the extrapolated sequence.
>
> [Q3.2] Differences between the proposed work and [17]?
>
> [A3.2] The goal of [17] is to learn alignment of two sequences. We adopt a similar way to find the alignment matrix between trimmed and full sequences. However, their task and our task are essentially different. We use those alignment matrices to augment the existing training data.
>
> [Q3.3] Eq. 4: k_i should be an index between 1 and T, but from the formula it does not seem so
>
> [A3.3] In Eq. 4, s_ij is the weight for each j and j is in the range [1,T], so the value of k_i is in  [1,T]. We have round(k_i) in line 179.
>
> [Q3.4] How to obtain the similarity matrix?
>
> [A3.4] See line 175-177 and Eq. (3).
>
> [Q3.5] Why clustering the W?
>
> [A3.5] (1) The original number of transforms for W is too large, note that n_W = n_training_samples x per_segment_in_one_sample. With clustering, we avoid inefficient sampling of W. (2) Important transform patterns (e.g. mirroring) can stand out during the clustering, as some of them only account for small percentage of the whole pool of W. We will add these explanation.
>
> [Q3.6] About the loss function.
>
> [A3.6] While it is certainly acceptable to have loss exactly as Eq 5, in practice, for efficiency, we randomly take batch_size x m_aug raw samples and batch_size x  (1-m_aug) augmented samples so that the total number of samples in a batch is still batch_size.
>
> 4. Experiments
>
> [Q4.1] Reporting the results with P as training.
>
> [A4.1] The result with P as training is provided in Table 3 in P51->N51 setting.
>
> [Q4.2] The transfer sub-settings are different from [41]. Why this choice?
>
> [A4.2] In Table 3 we follow exactly the same setting as [41] for fair comparison. However, [41] only considers mutual shared actions between two datasets. Our new multiple-dataset cross-dataset setting allows us to study and evaluate generalizability in a more fundamental way, so we mainly report results on our new setting (Table 1 and 2).
>
> [Q4.3] Kinetic never appears in the experiments
>
> [A4.3] Kinetics (K12) is used in Table 3.
>
> [Q4.4] How about the cross-view for NTU?
>
> [A4.4] We will discuss it briefly (see Reviewer UadT [Q1,A1]).
>
> [Q4.5] ERM?
>
> [A4.5] ERM refers to training a classifier using backbone network with standard  cross-entropy loss. It serves as a basis for all the methods.

---

> > ### Comment · Reviewer_Vnjk · 2024-08-12
> >
> > I truly thank the authors for the care they put in their rebuttal. My main concerns have been solved (some of my observations were actually due to misunderstandings on my side), and  I'm happy to raise my score. In any case, I suggest the authors to revise the introduction to make the story more clear (see my original comments).

---

> > > ### Author Response · Authors · 2024-08-13
> > >
> > > Dear Reviewer Vnjk:
> > >
> > > Thanks for the positive response and raising the score. We really appreciate your kind suggestions. In the revision for the introduction, we will add (1) clear insight/motivation for recovering complete actions as explained in the rebuttal, (2) a brief pipeline description for the full method, and (3) a clear justification for the need of clustering in each module as explained in the rebuttal. We will also address the issues for other parts of the paper as suggested.

---

### Official Review · Reviewer_V13L · 2024-07-11

**Soundness:** 2
**Presentation:** 3
**Contribution:** 2
**Rating:** 5
**Confidence:** 5

**Summary:**

This paper proposes a novel recover-and-resample augmentation framework to address the skeleton action generalization problem across different datasets. The framework utilizes a complete action prior to recover full action sequences from partial observations, employing boundary pose-conditioned extrapolation and smooth linear transforms. The proposed method demonstrates superior performance in cross-dataset settings compared to existing domain generalization approaches.

**Strengths:**

. Innovative Framework: The recover-and-resample augmentation framework is a novel approach to tackling the skeleton action generalization problem by focusing on recovering complete actions.

. Comprehensive Evaluation: The method is thoroughly validated across multiple datasets, showing significant improvement over baseline methods.

. Efficient Learning: The use of clustering for learning boundary poses and linear transforms makes the framework efficient and scalable.

. Powerful generalizability on cross datasets.

. Utilize nature concept of skeleton-based dataset for generalizability.

. Varied method suggestion: clustering concept and linear transformation algorithms

**Weaknesses:**

. Limited Scope: The evaluation is primarily focused on indoor datasets, which may limit the generalizability of the findings to other types of datasets or real-world applications.

. Complexity in Implementation: The two-step stochastic action completion and the need for clustering might pose implementation challenges for practitioners.

- Dependency on Clustering Algorithms: The approach heavily relies on clustering algorithms for learning boundary poses and linear transforms. The effectiveness of the method may be influenced by the choice of clustering algorithm and the parameters used, which might require extensive tuning and could impact the reproducibility and scalability of the approach.

- Request more ablation experiments on diverse sampling and clustering methods.

- Limited Exploration of Resampling Techniques: While the paper proposes a robust framework for recovering and resampling action sequences, the resampling techniques themselves are relatively simple (random) and not extensively explored. More advanced or varied resampling strategies could potentially further enhance the augmentation process and improve generalizability.

**Questions:**

No further questions.

**Limitations:**

Tackled in the draft.

---

> ### Author Rebuttal · Authors · 2024-08-07
>
> Dear Reviewer V13L:
>
> Thanks for your effort for reviewing our paper and giving kind suggestions. We really appreciate your positive comments on our paper.
>
> [Q1] Dependency on Clustering Algorithms: The approach heavily relies on clustering algorithms for learning boundary poses and linear transforms. The effectiveness of the method may be influenced by the choice of clustering algorithm and the parameters used, which might require extensive tuning and could impact the reproducibility and scalability of the approach.
>
> [A1] (1) We use k-means, which is one of the simplest clustering algorithms, and we conducted experiments for important parameters in the clustering process. (2) The insight behind is that we need to mine sufficient and important patterns for linear temporal transforms, e.g. scaling, shifting, reflection. So the number of linear transform clusters is important (Table 5), at least not to be set too small. (3) The number of clustered boundary poses is generally not very sensitive (Table 6).
>
> [Q2] Limited Exploration of Resampling Techniques: While the paper proposes a robust framework for recovering and resampling action sequences, the resampling techniques themselves are relatively simple (random) and not extensively explored. More advanced or varied resampling strategies could potentially further enhance the augmentation process and improve generalizability.
>
> [A2] (1) Yes, indeed we use very simple resampling method in our paper, namely randomly cropping a segment and then resampling it to remain the original sequence length. Despite its simplicity, it generally well addresses the case that many actions are only segments of their full sequences. (2) The whole recovering and resampling process already introduces some level of redundancy, since the recovering stage and resampling stage are both stochastic. (3) In limitation part we discussed the potential of more advanced resampling strategy in addressing ambiguity issue, but this seems to be a common challenge for all the augmentation-based methods. (4) Currently the resampling is done uniformly on randomly selected segments. We will try non-uniform sampling on random segments.  We really appreciate it if the reviewer is willing to offer some more insights on what kind of resampling method might improve the results.
>
> [Q3] Request more ablation experiments on diverse sampling and clustering methods.
>
> [A3] Thanks for the suggestion. In the revision we plan to add DBSCAN as another clustering algorithm to check the reproducibility, and add non-uniform resampling of random segments as another resampling method to see whether it can bring improvement.

---

### Official Review · Reviewer_UxEw · 2024-07-12

**Soundness:** 2
**Presentation:** 3
**Contribution:** 2
**Rating:** 4
**Confidence:** 5

**Summary:**

In this paper, the authors address the issue of generalizing skeleton-based action recognition across different domains. They propose a novel recover-and-resample augmentation framework based on the concept of complete action prior. The approach is validated on different cross-dataset settings and demonstrates a significant improvement in cross-dataset accuracy compared to existing methods.

**Strengths:**

1. Introduces a new recover-and-resample framework that effectively addresses temporal mismatch in skeleton action recognition.

2. Utilizes the concept of action completeness within large datasets, employing boundary poses and linear transforms to capture global action patterns.

3. The experimental results on different cross-dataset settings outperform the previous methods.

**Weaknesses:**

Although the method is intriguing and the performance is impressive, I am content with the current experimental setting. In real-world applications, dealing with cross-dataset issues makes it challenging to ensure that both datasets share the same action categories. Currently, the shared classes are manually selected. In my opinion, it would be better to adopt the new setting proposed by [1], where the source and target datasets have a category gap.

[1]   Collaborating Domain-shared and Target-specific Feature Clustering for Cross-domain 3D Action Recognition. ECCV 2022

**Questions:**

N.A.

**Limitations:**

N.A.

---

> ### Author Rebuttal · Authors · 2024-08-07
>
> Dear Reviewer UxEw:
>
> Thanks for your effort for reviewing our paper and giving kind suggestions.
>
> [Q1] I am content with the current experimental setting
>
> [A1] Do you mean you are not content with the current experimental setting? Since the generalizability of skeleton-based action recognition is still less explored in the community, we focus on the task of domain generalization, which is the most formal and standard setting where source and target datasets share the same action categories. Our reason for adopting this setting is that we can put every effort on investigating the domain gaps in skeleton-based action recognition in a more fundamental way. Following such a goal, we explored the inherent nature of action completeness for improving generalizability.
>
> In the new setting proposed by [1], where the source and target datasets have a category gap, useful information from large amount of unlabeled data can be mined. However, its final prediction is unable to give the actual action label in a straightforward way, especially for those unseen categories. Note that [1] needs to solve the label assignment problem to obtain the accuracy metric. It is more of a representation learning or a clustering problem. As a result, such a setting is also limited in its practical use.
>
> If considering another new setting where the source and target datasets have a category gap and the model has to predict the exact action labels for unseen target actions, it would become a zero-shot setting for unseen categories and more information about the category itself needs to be considered.
>
> So basically our standard setting and the new setting [1] address the generalization problem in different aspects. It is hard to say one setting is overwhelmingly better than the other. Hence, in our opinion, adopting a standard setting of domain generalization should not be blamed as core weakness of the paper.
>
>
> [1] Collaborating Domain-shared and Target-specific Feature Clustering for Cross-domain 3D Action Recognition. ECCV 2022

---

> > ### Comment · Reviewer_UxEw · 2024-08-14
> > **Rebuttal Response**
> >
> > Based on my understanding, in real-world applications, there's no certainty that newly collected video or skeleton sequences will correspond to classes that the model was originally trained on. Therefore, I contest the claim that "As a result, such a setting is also limited in its practical use." This introduces a significant challenge in real-world contexts, necessitating further research and development to tackle it effectively.
> >
> > Given that the main title of this paper is "Cross-Dataset Skeleton Action Recognition," as opposed to domain generalization in skeleton action recognition, my primary concern is with the accuracy of the cross-dataset setting definition. In my view, we should not align labels or categories before proceeding with cross-dataset experiments.
> >
> > In conclusion, I will maintain my rating at '4: Borderline reject'.

---

### Decision · Program_Chairs · 2024-09-25

**Decision:**

Accept (poster)

**Comment:**

3 of 4 reviewers were inclined to accept the paper with one reviewer flipping to positive. The negative review had very limited comments and did not respond to the rebuttal. The AC has read all reviews and agrees with the majority of reviewers that the paper should be accepted.